# A 24-month National Cohort Study examining long-term effects of COVID-19 in children and young people

Terence Stephenson [1,15], Snehal M. Pinto Pereira [2,15] ✉, Manjula D. Nugawela[1], Emma Dalrymple[1], Anthony Harnden[3], Elizabeth Whittaker[4], Isobel Heyman[1], Tamsin Ford[5], Terry Segal[6], Trudie Chalder[7], Shamez N. Ladhani[8], Kelsey McOwat[8], Ruth Simmons[8], Laila Xu[1], Lana Fox-Smith [1], CLoCk Consortium* & Roz Shafran[1]

## Abstract

**Background** Some children and young people (CYP) infected with SARS-COV-2 experience impairing symptoms post-infection, known as post-COVID-19 condition (PCC). Using data from the National Long COVID in Children and Young People (CloCk) study, we report symptoms and their impact up to 24-months post-infection. **Methods** CloCk is a cohort of CYP in England aged 11-to-17-years when they had a SARS-CoV-2 PCR-test (between September 2020 and March 2021). Of 31,012 eligible CYP 24-months post-PCR test, 12,632 participated (response = 40.7%). CYP were grouped by infection status: 'initial test-negatives; no subsequent positive-test' (NN); 'initial test-negatives; subsequent positive-test' (NP); 'initial test-positives; no reported re-infection' (PN); and 'initial test-positives; reported re-infection' (PP). The Delphi research definition of PCC in CYP was operationalised; symptom severity/impact and validated scales (e.g., Chalder Fatigue Scale) were recorded. We examine symptom profiles 24-month post-index-test by infection status. **Results** 7.2% of CYP consistently fulfil the PCC definition at 3-, 6-, 12- and 24-months. These CYPs have a median of 5-to-6 symptoms at each time-point. Between 20% and 25% of all infection status groups report 3+ symptoms 24-months post-testing; 10–25% experience 5+ symptoms. The reinfected group has more symptoms than the other positive groups; the NN group has the lowest symptom burden ($p < 0.001$). PCC is more common in older CYPs and in the most deprived. Symptom severity/impact is higher in those fulfilling the PCC definition. **Conclusions** The discrepancy in the proportion of CYP fulfilling the Delphi PCC definition at 24-months and those consistently fulfilling the definition across time, highlights the importance of longitudinal studies and the need to consider clinical impairment and range of symptoms.

## Plain language summary

Some children and young people infected with SARS-COV-2 experience impairing symptoms long after infection; this is known as 'Long COVID'. We used data from the Long COVID in Children and Young People (CloCk) study to describe symptoms and how much they impact children and young people's lives 24-months post-infection. We found that 7.2% of children and young people consistently meet the 'Long COVID' research definition at 3-, 6-, 12- and 24-months post-infection. These children and young people reported around 5-to-6 symptoms at each time-point. Reinfected children and young people had more symptoms than children and young people who report one infection; those who report no infection had the lowest symptom burden. When researching Long COVID, we need to consider clinical impairment and the range of symptoms reported.

Post-COVID-19 condition (PCC), also called long COVID, in children and young people (CYP) under 18 years old is a difficult area of research, with many challenges compared to other classical epidemiological studies[1,2]. In the absence of a consistent surrogate outcome, proxy biomarker, or specific imaging finding for PCC, early in the pandemic, Delphi consensus methodology provided a research definition of PCC[3]. However, relatively few

research studies on long COVID use any definition, further increasing the challenge of conducting research in this area[4]. Additional complications are that many people were infected with SARS-CoV-2 before diagnostic tests were developed; most symptoms (other than loss of taste and smell) are not condition-specific; and there is variability in time and frequency of vaccinations in relation to SARS-CoV-2 infection. As the pandemic progressed,

A full list of affiliations appears at the end of the paper. *A list of authors and their affiliations appears at the end of the paper. ✉e-mail: snehal.pereira@ucl.ac.uk

repeated waves of new SARS-CoV-2 variants and mass infection made it impossible to compare infected young people to an uninfected group. Finally, the general disruption to the lives of CYP due to the COVID-19 pandemic may contribute to somatic symptomatology, as may a chronic post-viral syndrome[5].

A recent systematic review emphasised that current evidence is dominated by smaller and often single-site studies, frequently without using a recognised PCC definition and with short follow-up[6,7]. It is important to follow-up young people longitudinally to understand the natural course of symptoms after SARS-CoV-2 infection due to the impact on both public health and clinical care. One large-scale, national study that operationalises a published PCC Delphi research definition is the Long COVID in Children and Young People (CLoCk) study. This study has reported findings on PCC in 20,202 CYP living in England up to 12-month after their initial PCR test[8,9].

In previous CLoCk publications, test-positive CYP were compared to laboratory-confirmed SARS-CoV-2 test-negative CYP. However, according to the COVID-19 Schools Infection Survey in England, reporting in June 2022 on data from March 2022, 99% of teenagers had SARS-CoV-2 antibodies, consistent with prior infection and/or COVID-19 vaccination[10]. Of

the 99.0% of secondary school pupils with SARS-CoV-2 antibodies, 65% had been vaccinated, and 34% were unvaccinated. Vaccinated pupils could develop antibodies through vaccination and/or natural infection, whereas unvaccinated pupils will only have antibodies following natural infection. Although it is possible in principle to differentiate between the antibodies from infection and vaccination status with specific testing, it was not possible within this study. Therefore, it is now no longer possible to compare test-positive with test-negative CYP from the CLoCk study. Instead, here we report on symptoms self-reported 24-months post-testing on over 12,600 CYP categorised into four groups by infection status over the 24-month period, including a group that never tested positive (which includes true 'negatives' and/or undetected infections). By operationalising the Delphi definition of PCC up to 24-months post-PCR-testing, we are able to address questions about the profile, persistence, and impact of symptoms post-infection.

Specifically, we hypothesised that: (1) members of all four infection status groups of CYP would have some symptoms 24-months after initial testing; (2) the most common symptoms would include some of the tiredness, sleeping difficulties, shortness of breath, abdominal pain, con-

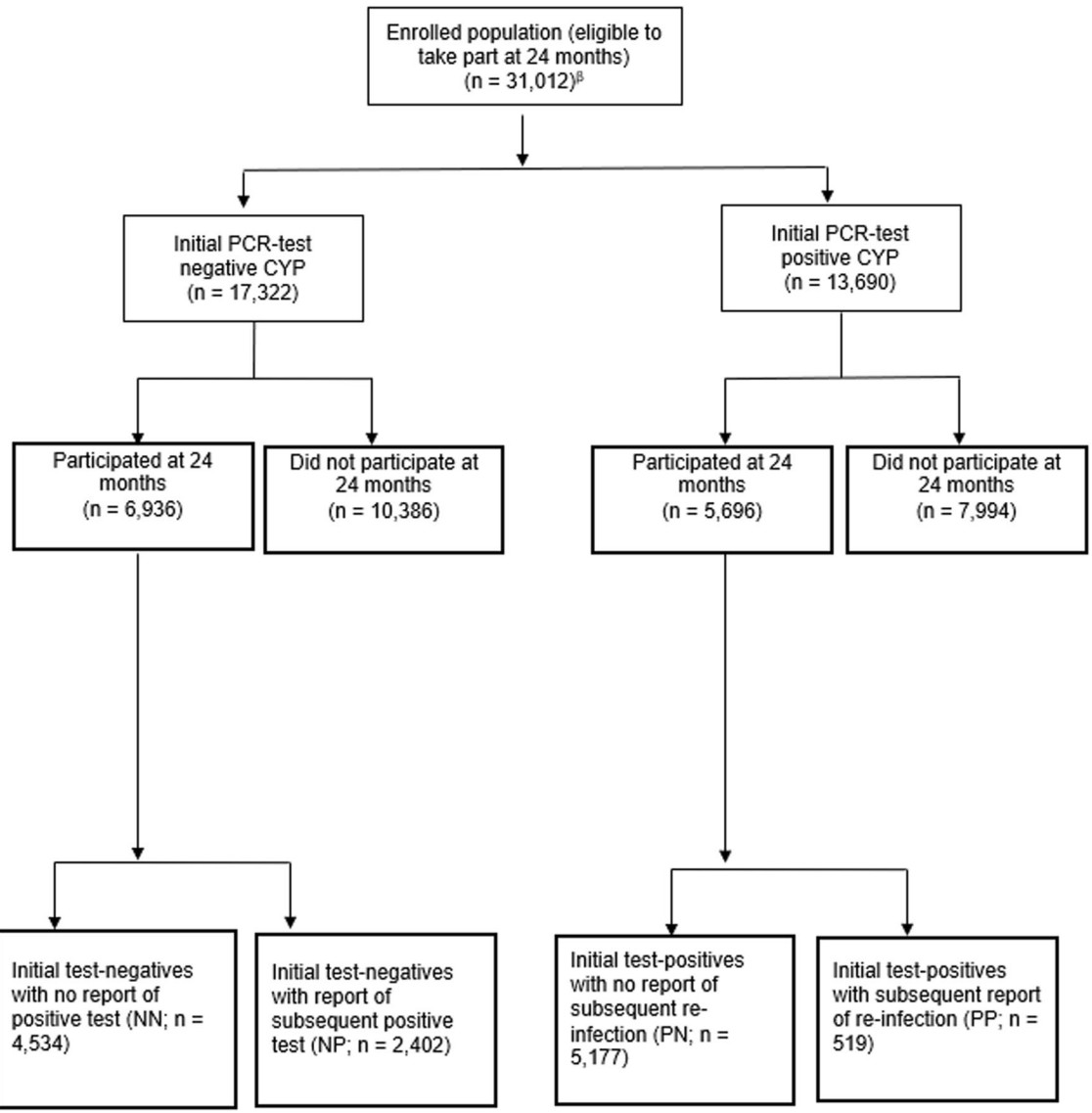

**Fig. 1 | Participant flow diagram[¥].** [¥]Classification into the four groups accounts for PCR test information held at UKHSA and also self-report. [β]Enrolled population considers only children and young people who were invited to respond at 24-month, i.e. those who registered at 3-month (n = 7356), 6-month (n = 10,530) or 12-month (n = 13,126).

**Table 1 | Demographics (N(%)) of the target population, non-responders, and responders included in the 24-month sample**

| | Test-negatives: invited (N = 17,322) | Test-negative: non-responders (N = 10,386) | Test-negative: responders (N = 6936) Participating initial-negatives (NN; N = 4534) | Participating initial-negatives & infected (NP; N = 2402) | p[a] | Test-positives: invited (N = 13,690) | Test-positive: non-responders (N = 7994) | Test-positive: responders (N = 5696) Participating initial-positives (PN; N = 5177) | Participating initial-positive & reinfected (PP; N = 519) | p[a] |
|---|---|---|---|---|---|---|---|---|---|---|
| | | | Response rate: 40.0% | | | | | Response rate: 41.6% | | |
| *Sex* | | | | | | | | | | |
| Female | 10,666 (61.6) | 6112 (58.9) | 2974 (65.6) | 1580 (65.8) | <0.001 | 8385 (61.3) | 4709 (58.9) | 3337 (64.5) | 339 (65.3) | <0.001 |
| Male | 6656 (38.4) | 4274 (41.2) | 1560 (34.4) | 822 (34.2) | | 5305 (38.7) | 3285 (41.1) | 1840 (35.5) | 180 (34.7) | |
| *Age at index test (years)* | | | | | | | | | | |
| 11–14 | 8459 (48.8) | 5302 (51.1) | 1891 (41.7) | 1266 (52.7) | <0.001 | 6398 (46.7) | 3795 (47.5) | 2342 (45.2) | 261 (50.3) | 0.04 |
| 15–17 | 8863 (51.2) | 5084 (49.0) | 2643 (58.3) | 1136 (47.3) | | 7292 (53.3) | 4199 (52.5) | 2835 (54.8) | 258 (49.7) | |
| *Ethnicity* | | | | | | | | | | |
| White | 13,043 (75.3) | 7763 (74.7) | 3333 (73.5) | 1947 (81.1) | 0.03 | 10,155 (74.2) | 5900 (73.8) | 3856 (74.5) | 399 (76.9) | 0.16 |
| Asian/Asian British | 2441 (14.1) | 1483 (14.3) | 720 (15.9) | 238 (9.9) | | 2112 (15.4) | 1247 (15.6) | 789 (15.2) | 76 (14.6) | |
| Mixed | 934 (5.4) | 565 (5.4) | 241 (5.3) | 128 (5.3) | | 681 (5.0) | 390 (4.9) | 263 (5.1) | 28 (5.4) | |
| Black/African/Caribbean | 543 (3.1) | 328 (3.2) | 158 (3.5) | 57 (2.4) | | 390 (2.9) | 230 (2.9) | 151 (2.9) | 9 (1.7) | |
| Other | 260 (1.5) | 180 (1.7) | 53 (1.2) | 27 (1.1) | | 264 (1.9) | 175 (2.2) | 87 (1.7) | 2 (0.4) | |
| Prefer not to say | 101 (0.6) | 67 (0.7) | 29 (0.6) | 5 (0.2) | | 88 (0.6) | 52 (0.7) | 31 (0.6) | 5 (1.0) | |
| *Region* | | | | | | | | | | |
| East Midlands | 1159 (6.7) | 667 (6.4) | 315 (7.0) | 177 (7.4) | 0.001 | 1051 (7.7) | 613 (7.7) | 396 (7.6) | 42 (8.1) | 0.004 |
| East of England | 3836 (22.2) | 2238 (21.6) | 989 (21.8) | 609 (25.4) | | 2211 (16.2) | 1260 (15.8) | 878 (17.0) | 73 (14.1) | |
| London | 3539 (20.4) | 2161 (20.8) | 961 (21.2) | 417 (17.4) | | 2,618 (19.1) | 1537 (19.2) | 988 (19.1) | 93 (17.9) | |
| North East England | 602 (3.5) | 375 (3.6) | 160 (3.5) | 67 (2.8) | | 596 (4.3) | 368 (4.6) | 210 (4.1) | 18 (3.5) | |
| North West England | 1895 (10.9) | 1180 (11.4) | 495 (10.9) | 220 (9.2) | | 1711 (12.5) | 1044 (13.1) | 600 (11.6) | 67 (12.9) | |
| South East England | 2750 (15.9) | 1588 (15.3) | 735 (16.2) | 427 (17.8) | | 2167 (15.8) | 1201 (15.0) | 883 (17.1) | 83 (16.0) | |
| South West England | 772 (4.5) | 453 (4.4) | 192 (4.2) | 127 (5.3) | | 742 (5.4) | 414 (5.2) | 284 (5.5) | 44 (8.5) | |
| West Midlands | 1601 (9.2) | 999 (9.6) | 403 (8.9) | 199 (8.3) | | 1431 (10.4) | 862 (10.8) | 519 (10.0) | 50 (9.6) | |
| Yorkshire and the Humber | 1168 (6.7) | 725 (7.0) | 284 (6.3) | 159 (6.6) | | 1163 (8.5) | 695 (8.7) | 419 (8.0) | 49 (9.4) | |
| *IMD quintile*[b] | | | | | | | | | | |
| 1 (most deprived) | 2929 (16.9) | 1836 (17.7) | 758 (16.7) | 335 (14.0) | <0.001 | 2416 (17.6) | 1483 (18.6) | 825 (16.0) | 108 (20.8) | 0.001 |
| 2 | 3071 (17.7) | 1873 (18.0) | 815 (18.0) | 383 (16.0) | | 2477 (18.1) | 1458 (18.2) | 928 (17.9) | 91 (17.6) | |
| 3 | 3283 (18.9) | 1964 (18.9) | 859 (19.0) | 460 (19.2) | | 2509 (18.3) | 1470 (18.4) | 941 (18.2) | 98 (18.9) | |
| 4 | 3739 (21.6) | 2248 (21.6) | 965 (21.3) | 526 (21.9) | | 2917 (21.3) | 1704 (21.3) | 1108 (21.4) | 105 (20.2) | |

**Table 1 (continued) | Demographics (N(%)) of the target population, non-responders, and responders included in the 24-month sample**

| | Test-negatives: invited (N = 17,322) | Test-negative: non-responders (N = 10,386) | Test-negative: responders (N = 6936) | | p^a | Test-positives: invited (N = 13,690) | Test-positive: non-responders (N = 7994) | Test-positive: responders (N = 5696) | | p^a |
|---|---|---|---|---|---|---|---|---|---|---|
| | | | Participating initial-negatives (NN; N = 4534) | Participating initial-negatives & infected (NP; N = 2402) | | | | Participating initial-positives (PN; N = 5177) | Participating initial-positive & reinfected (PP; N = 519) | |
| 5 (least deprived) | 4300 (24.0) | 2465 (23.7) | 1137 (25.1) | 698 (29.1) | | 3371 (24.6) | 1879 (23.5) | 1375 (26.5) | 117 (22.5) | |

Data are rounded to 1 decimal point, percentages may sometimes add up to just above (100.1) or below (99.9) 100%.

^a p-values from chi-squared tests comparing test-negative non-responders vs. test-negative responders and similarly test-positive non-responders vs. test-positive responders

^b IMD: index of multiple deprivation. The index of multiple deprivation (IMD) was calculated from the CYP's small local area level-based geographic hierarchy (lower super output area) at the time of the first questionnaire and used as a proxy for socio-economic status. We report IMD quintiles from most (quintile 1) to least (quintile 5) deprived.

centration difficulties, muscle pain and headaches as identified in previous PCC studies[6,11,12]; (3) older CYP, females, and CYP from ethnic minorities will have a higher PCC prevalence; (4) CYP that were infected multiple times would have more symptoms and with greater impact than CYP that tested positive once and; (5) symptoms would not differ by vaccination status, as identified in our previous PCC studies[8]. We show that 7.2% of CYP consistently meet the research definition of PCC at 3-, 6-, 12- and 24-months post-infection. These CYPs report 5-to-6 symptoms at each time-point. We find that reinfected CYP had more symptoms than CYP who reported one infection; those who reported no infection had the lowest symptom burden.

## Methods

The CLoCk study, described in detail elsewhere, is a cohort study of SARS-CoV-2 PCR-positive CYP[13]. Briefly, CYP who PCR-tested positive between September 2020 and March 2021 when they were aged 11-to-17-years, were matched at study invitation on month of test, age, sex at birth, and geographical area to SARS-CoV-2 test-negative CYP using the SARS-CoV-2 testing dataset held by the United Kingdom Health Security Agency (UKHSA). After obtaining written informed consent, CYP completed an online questionnaire about their demographics (e.g., ethnicity) and health at the time of their SARS-CoV-2 PCR test ("baseline"; retrospectively reported) and contemporaneously at approximately 3, 6, 12 and 24 months after their index-PCR test (with different numbers of respondents at each time point depending on the time of recruitment into the study, relative to their test date; Supplementary Fig. 1). Of note, the index-PCR test for all young people occurred prior to vaccine rollout in August 2021.

The dataset held by UKHSA can track repeated PCR testing longitudinally within individuals. In addition, at each contact, we asked CYP if they ever tested positive for SARS-CoV-2 (thereby allowing CYP to report on any PCR or Lateral Flow Test taken). Using information on subsequent testing held by UKHSA and from self-report, we divided the CYP into four groups (detailed below) and in Fig. 1.

The four groups of CYP were: 'initial test-negatives with no subsequent positive test' (negative–negative; NN); 'initial test-negatives with a subsequent positive test' (negative–positive; NP); 'initial test-positives with no report of subsequent re-infection' (positive–negative; PN); and 'initial test-positives with a report of subsequent re-infection' (positive–positive; PP).

### Measures

The CLoCk questionnaire included demographics, elements of the International Severe Acute Respiratory and Emerging Infection Consortium (ISARIC) Paediatric COVID-19 questionnaire, the recent Mental Health of Children and Young People in England surveys and originally included 21 symptoms (mostly assessed as present/absent)[6,11,12]. Validated health scales: the Strengths and Difficulties Questionnaire (SDQ)[14], Short Warwick Edinburgh Mental Wellbeing Scale (SWEMWS)[10], Chalder Fatigue Scale[15], and the EQ-5D-Y[16] (as a measure of quality of life and function), were also included. The CLoCk questionnaire was largely unchanged between study enrolment and subsequent follow-ups: redundant questions (e.g., demographics, symptoms at the time of testing, etc.) were removed at follow-ups, and a number of new questions were added (e.g., sleeping difficulties, a symptom severity scale, a symptom impact scale, and questions on the type of COVID-19 vaccination received and date each dose was administered) by the 24-month data collection sweep[13].

The Delphi research definition of PCC in CYP[3] was operationalised at the time of questionnaire completion (i.e., 24-months post-index test) as experiencing ≥1 symptom AND problems with at least one of: mobility OR self-care OR doing usual activities OR having pain/discomfort OR feeling very worried/sad, based on the EQ-5D-Y scale. CYP meeting this operationalised research definition at 24-month post-index test were classified as having PCC at that time. Vaccination status (dosage: 0, 1, 2, 3+) was determined from information held at UKHSA; when information was missing (n = 361), self-reported information from the 24-month post-index test questionnaire was used.

## Statistical methods

To assess the representativeness of study participants, we compared their demographic characteristics (sex, age, ethnicity, region of residence, and Index of Multiple Deprivation) to the population invited to take part at 24 months by infection status. To describe the symptom profiles in all four groups 24 months after initial testing, we tabulated the total number of symptoms (0, 1, 2, 3, 4, and 5+) reported, the prevalence of individual symptoms, self-rated health, and symptom severity, and impact 24 months post index-test. We also examined the prevalence of individual symptoms graphically. We assessed the prevalence of meeting the PCC research definition 24 months post-index-test by infection status. For the NN group the requirement of a positive test was excluded. For the NP and PP groups, we were unable to always confirm with certainty the last date of a positive test (because CYP were able to self-report having a positive test but not the date of the positive test). Hence, for these two groups, we assume that their last infection was over 3 months prior to the 24 month questionnaire. We examined study participant characteristics and 24 months post-index-test health and well-being (via validated scales) stratified by infection status and PCC status at 24 months. We used chi-squared or Mann–Whitney tests to determine if symptom profiles at 24 months post index-test differed by infection status. Finally, study participant characteristics on validated scales 24 months post index-test were stratified by infection status groups and vaccination status (dosage: 0, 1, 2, 3+) at 24 months. Most questions in the CLoCk questionnaire were compulsory, so missing data was minimal. However, questions on symptom severity and impact were added part-way through 24-month data collection, so only available for a sub-sample. Questions regarding vaccination were optional and, therefore, had missing data. In this case, vaccination data was primarily used from UKHSA databases and supplemented with self-report: three CYPs were excluded when examining vaccination status because their vaccination data was not available from UKHSA databases or self-report.

Data management and analysis were performed using STATA16.0.

## Consent, ethics, and registration details

Informed consent was obtained for CYP aged 16 or older; assent and parental informed consent were obtained for under 16-year-olds. Ethical approval was provided by the Health Research Authority Yorkshire and the Humber—South Yorkshire Research Ethics Committee (REC reference: 21/YH/0060; IRAS project ID: 293495). This study is registered with the ISRCTN registry (ISRCTN 34804192).

## Results

### Analytical sample

Of the 31,012 CYP invited to fill in a questionnaire 24 months post-PCR test, 12,632 CYP participated and were included in our analytic sample (response rate = 40.7%; Fig. 1 and Supplementary Fig. 1). At 24 months post-PCR test, participating CYP were generally more likely to be females, older and from lesser deprived areas compared to non-responders (Table 1).

### 24-month follow-up

24-month follow-up questionnaires were returned at a median of 104.1 weeks [IQR:103.0, 105.6] after the index-PCR test. Prevalence of at least one dose of the COVID-19 vaccination by 24 months varied from 90% (4079/4531) for the NN group to 78.2% (406/519) for the PP group (Supplementary Table 1). For the PP and NP groups, we are unable to determine the chronology of (re)infection and vaccination due to the self-report nature of the questionnaire.

At 24 months post index-test, all infection status groups reported symptoms, with tiredness, trouble sleeping, shortness of breath and headaches being most prevalent (Table 2, Fig. 2). However, symptom profiles differed by infection status. In general, the NN group had the lowest prevalence of symptoms, with the NP and PN groups having a higher prevalence and the PP group having the highest prevalence. For example, for tiredness, the prevalence was 38% (1736/4534) in the NN group, ~45% in the NP (1070/2402) and PN (2336/5177) groups, and 48% (249/519) in the

PP group ($p < 0.001$). Due to variation in prevalence of reported symptoms by infection status, the total number of reported symptoms also varied by infection status. For example, no symptoms ranged from 35·5% (184/519) in the PP group to 46·2% (2,093/4,534) in the NN group, while 5+ symptoms were reported by 14·2% (643/4,534) of NN to 20.8% (108/519) of PP CYP ($p < 0.001$). Similarly, GP consultations over the 24-month period in relation to COVID-19 varied from 7.7% (348/4534) in the NN group to 14.6% (76/519) in the PP group. In contrast, there was little difference in self-rated overall health, symptom severity, or symptom impact at 24 months by infection status ($p > 0.10$; Table 2).

Approximately one-quarter of CYP fulfilled the broad Delphi research definition of PCC 24 months post index-testing in the NN, PN, and NP groups (Table 3); while 30% of the PP group met this broad PCC definition 24 months after their first infection. PCC was more common in older than younger CYPs and in the most, compared to the least, deprived quintile. Strikingly, PCC was almost twice as common in females compared to males in all infection status groups, and there was little difference by ethnicity. At 24 months post-index-test, for those fulfilling the PCC Delphi research definition, SDQ scores were higher, indicating more difficulties, SWEMBS were lower, indicating worse quality of life, and Chalder fatigue scale scores were higher, indicating more tiredness compared to those not meeting the PCC Delphi research definition (Table 4). Likewise, self-rated health was lower, while symptom severity and symptom impact were higher in those fulfilling the PCC Delphi research definition, irrespective of infection history.

### PCC in index-positive CYP followed-up 3–24 months

When examining the sub-set of 943 index-test positive CYP with data at 3, 6, 12 and 24 months post index-test longitudinally, 233 (24.7% of 943) fulfilled the PCC Delphi research definition at 3 months follow-up, 135 (14.3%) continued to fulfil the PCC Delphi research definition at 6 months, and 94 (10.0%) continued to fulfil the PCC Delphi research definition at 12 months. Only 68 of these 943 CYP (7.2%) continued to fulfil the PCC Delphi research definition at the 24-month follow-up. Therefore, 7.2% consistently fulfilled the PCC Delphi research definition at 3, 6, 12, and 24 months (Fig. 3). These 68 CYP reported a median of 5 symptoms (IQR 3,6) at 3 months, 5 symptoms (IQR 3,7) at 6-months, 6 symptoms (IQR 4, 8.75) at 12 months and 5 symptoms (IQR 3.25, 8) at 24 months post-index testing.

### Vaccination uptake

Supplementary Table 1 shows that across all four infection status groups, vaccination uptake was lower for younger compared to older CYP. In general, white CYPs were more likely to be vaccinated compared to non-white CYPs (especially Black African or Caribbean). However, much more striking is that across groups, CYP in the most deprived quintile were three times more likely to be unvaccinated compared to those in the least deprived quintile.

Supplementary Table 2 shows that irrespective of infection status, vaccinated (dosage: 1, 2 or 3+) versus unvaccinated (dosage: 0), there were no obvious trends in the number of symptoms reported, health, well-being, or symptom impact or severity at 24 months.

## Discussion

Consistent with our hypotheses, a proportion of all four groups of CYP reported symptoms 24 months after initial testing, and the most common symptoms were tiredness, trouble sleeping, shortness of breath, and headaches. While tiredness was the most prevalent symptom across all infection groups, the prevalence of abdominal pain, concentration difficulties (i.e., confusion, disorientation, or drowsiness), and muscle pain was relatively low. Although 25–30% of all groups met the PCC Delphi research definition at 24 months post-index-test, only 7.2% of index-test positives met the criteria at 3, 6, 12, and 24 months post-index-test. This group consistently had a median of 5 or 6 symptoms and was, therefore, similar to the profile of young people seeking treatment for PCC from clinical services, the majority of whom had 5 or more symptoms at the time of seeking help[17].

**Table 2 | Reported symptoms N (%), self-rated health[a], symptom severity[a] and impact[a] by SARS-CoV-2 status 24 months post-test**

| | 24 months post-test | | | | |
|---|---|---|---|---|---|
| | Initial-negatives (NN) n = 4534 | Negative & infected (NP) n = 2402 | Initial-positives (PN) n = 5177 | Positive & re-infected (PP) n = 519 | p value[b] |
| *Number of reported symptoms* | | | | | |
| 0 | 2093 (46.2) | 967 (40.3) | 1969 (38.0) | 184 (35.5) | <0.001 |
| 1 | 724 (16.0) | 406 (16.9) | 886 (17.1) | 96 (18.5) | |
| 2 | 481 (10.6) | 272 (11.3) | 651 (12.6) | 50 (9.6) | |
| 3 | 332 (7.3) | 183 (7.6) | 456 (8.8) | 41 (7.9) | |
| 4 | 261 (5.8) | 149 (6.2) | 362 (7.0) | 40 (7.7) | |
| ≥5 | 643 (14.2) | 425 (17.7) | 853 (16.5) | 108 (20.8) | |
| *Specific symptoms* | | | | | |
| Fever | 156 (3.4) | 102 (4.2) | 201 (3.9) | 19 (3.7) | 0.386 |
| Chills | 475 (10.5) | 269 (11.2) | 620 (12.0) | 71 (13.8) | 0.040 |
| Persistent cough | 517 (11.4) | 294 (12.2) | 607 (11.7) | 64 (12.3) | 0.740 |
| Tiredness | 1736 (38.3) | 1070 (44.6) | 2336 (45.1) | 249 (48.0) | <0.001 |
| Shortness of breath | 838 (18.5) | 526 (21.9) | 1201 (23.2) | 128 (24.7) | <0.001 |
| Loss of smell | 112 (2.5) | 160 (6.7) | 274 (5.3) | 38 (7.3) | <0.001 |
| Unusually hoarse voice | 143 (3.2) | 97 (4.0) | 172 (3.3) | 13 (2.5) | 0.159 |
| Unusual chest pain | 246 (5.4) | 156 (6.5) | 366 (7.1) | 49 (9.4) | <0.001 |
| Unusual abdominal pain | 156 (3.4) | 101 (4.2) | 205 (4.0) | 25 (4.8) | 0.227 |
| Diarrhoea | 108 (2.4) | 53 (2.2) | 129 (2.5) | 17 (3.3) | 0.532 |
| Headaches | 652 (14.4) | 379 (15.8) | 832 (16.1) | 106 (20.4) | 0.002 |
| Confusion, disorientation or drowsiness | 191 (4.2) | 104 (4.3) | 276 (5.3) | 42 (8.1) | <0.001 |
| Unusual eye-soreness | 213 (4.7) | 109 (4.5) | 260 (5.0) | 26 (5.0) | 0.785 |
| Skipping meals | 342 (7.5) | 192 (8.0) | 434 (8.4) | 52 (10.0) | 0.168 |
| Dizziness or light-headedness | 416 (9.2) | 257 (10.7) | 602 (11.6) | 73 (14.1) | <0.001 |
| Sore throat | 495 (10.9) | 283 (11.8) | 579 (11.2) | 50 (9.63) | 0.489 |
| Unusual strong muscle pains | 119 (2.6) | 110 (4.6) | 225 (4.4) | 39 (7.5) | <0.001 |
| Earache or ringing in ears | 240 (5.3) | 150 (6.2) | 338 (6.5) | 44 (8.5) | 0.007 |
| Raised welts on skin or swelling | 44 (1.0) | 23 (1.0) | 62 (1.2) | 5 (1.0) | 0.667 |
| Red/purple sores/blisters on feet | 26 (0.6) | 24 (1.0) | 54 (1.0) | 3 (0.6) | 0.057 |
| Sleeping difficulties | 1067 (23.5) | 654 (27.2) | 1406 (27.2) | 172 (33.1) | <0.001 |
| Other | 175 (3.9) | 122 (5.1) | 234 (4.5) | 25 (4.8) | 0.104 |
| Did you/your parent talk to the doctor about your Covid-19 symptoms? | 348 (7.7) | 249 (10.4) | 468 (9.0) | 76 (14.6) | <0.001 |
| Did you go to hospital about your Covid-19? | 102 (2.3) | 73 (3.0) | 154 (3.0) | 23 (4.4) | 0.01 |
| Self-rated health[a] | 85 (70, 95) | 85 (70, 90) | 85 (70, 90) | 85 (70, 90) | 0.888 |
| Symptom severity[a,c] | 50 (30, 70) | 50 (30,60) | 50 (30, 70) | 50 (30, 65) | 0.645 |
| Symptom impact[a,d] | 40 (20, 70) | 50 (20, 70) | 40 (20, 70) | 40 (20, 70) | 0.104 |

[a]Reported as median(IQR), scored on a scale of 0 (worst) to 100 (best) for self-rated health; 0 (not severe at all) to 100 (extremely severe) for symptom severity; 0 (no impact) to 100 (extreme impact) for symptom impact.
[b]p value from chi-squared test of association between SARS-CoV-2 status 24 months post index-test and number of symptoms/specific symptoms and from Mann-Whitney tests between SARS-CoV-2 status 24 months post index-test and self-rated health, symptom severity and symptom impact.
[c]Based on sub-sample N = 4977.
[d]Based on sub-sample N = 4972.

Older CYP and females were more likely to fulfil the PCC Delphi research definition 24 months post-testing. Although all ethnicities were equally likely to fulfil PCC, it was noticeable that non-white CYP were much less likely to be vaccinated. The reinfected (PP) group had more symptoms than the other two positive groups (i.e., NP, PN) and thus had a higher prevalence of PCC (30% vs. ~25%). In addition, the PP group had more GP consultations related to COVID-19 and hospital attendance compared to the NP and PN groups, acknowledging the percentage of such consultations and attendances was small. We did not find that symptoms or their impact differed by vaccination status. Importantly, there was no significant difference in self-rated overall health, symptom severity, or symptom impact at 24 months by infection status.

The symptoms reported are comparable with other studies in young people in terms of prevalence and symptom range[18,19], although, to our

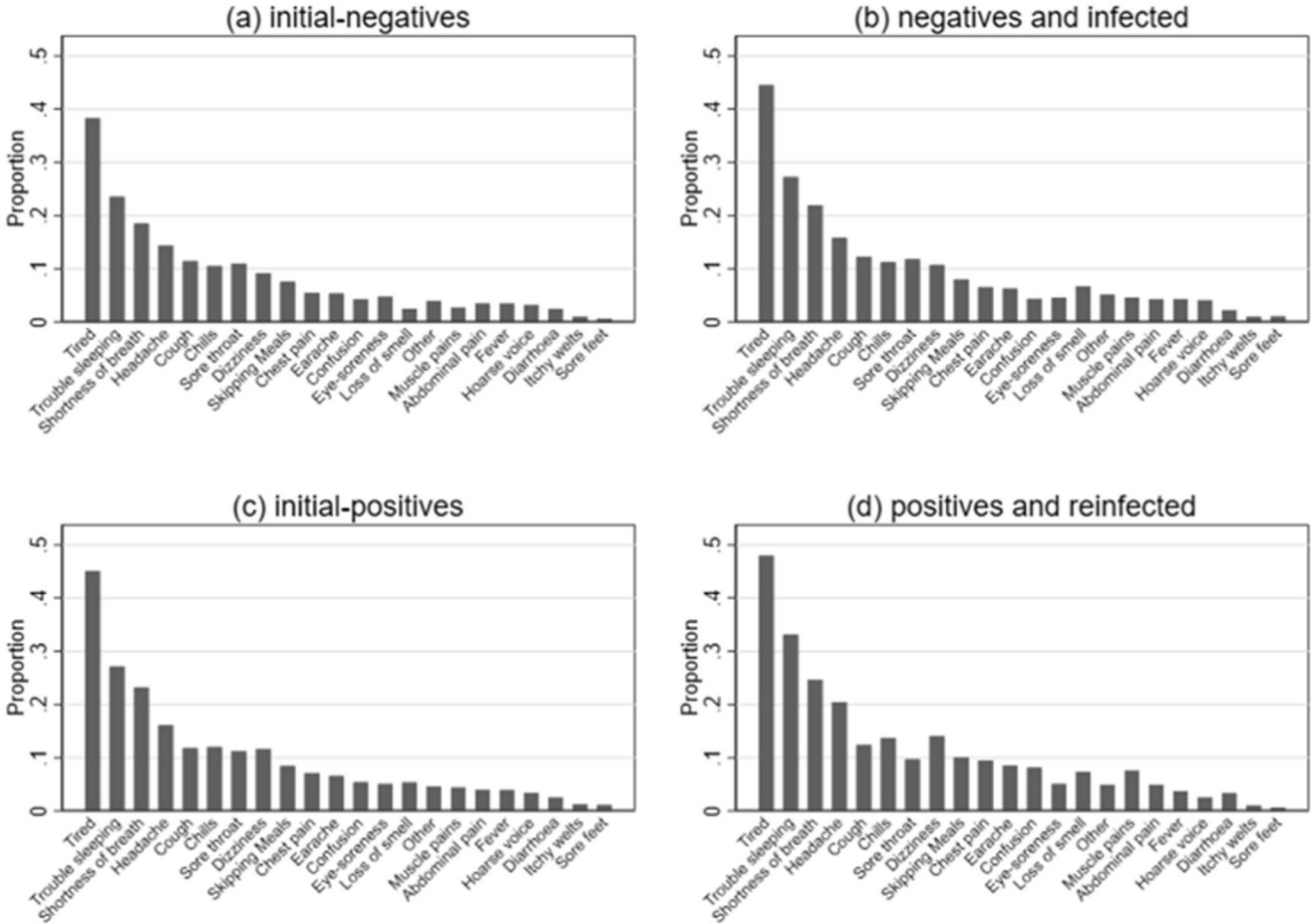

**Fig. 2 | Proportion\* of participants reporting symptoms 24 months post-testing.** Proportions shown among **a** initial-negatives (*n* = 4534), **b** negatives and infected (*n* = 2402), **c** initial-positives (*n* = 5177), and **d** positives and reinfected (*n* = 519). \*See Table 2 for exact values

knowledge, no previous study has reported symptoms at 24 months post-infection. The most common symptoms in adults, which have been reported at 24-month post-infection are shortness of breath, dizziness, heart racing, headaches, back pain, chest pain, fatigue, and trouble sleeping[20]. These findings are similar to our current CYP study: at 24-month follow-up, the most common symptoms were tiredness, trouble sleeping, shortness of breath, and headache with overall symptom prevalence generally higher in those with recurrent SARS-CoV-2 infection compared to those who never tested positive for the virus. In adults, the veterans' study showed that the population that exhibited a second infection was different from the population that experienced a single infection. However, methodological concerns remain, and thus we are cautious about overstating claims regarding re-infection in individual patients[21].

We found that 5+ symptoms were reported by 14.2% of those who never tested positive for SARS-CoV-2 and 20.8% of those with at least two infections. This is similar to other CYP studies of PCC reporting post-infection symptoms even after mild COVID-19[7]. In a large, recent cross-sectional study of CYP[18], 13.3% of 12-to-17-year-olds with prior symptomatic SARS-CoV-2 infection had persistent symptoms for 3-months or more following COVID-19, mainly loss/change of sense of smell (52.2%) and taste (40.7%). Additionally, one in nine of these children reported a large impact on their ability to carry out day-to-day activities due to their symptoms, which is broadly similar to the overall proportion in the CLoCk study seeking help from their GP, albeit, by 24 months follow-up.

Across all 4 sub-groups, self-reported symptoms and health at 24-months were of a similar prevalence to those reported by us at 12-months[8]. For example, we previously reported that 10% of NN, 16–20% of PNs and NPs, and 24% of PPs had 5+ symptoms 12 months post index-test; at

24 months corresponding values were 14% (NN), 17-18% (PN and NP) and 21% (PP)[8]. Essentially, the data show that at 12 and 24 months follow-up, between one-tenth and a quarter of CYPs experience 5+ symptoms, depending on which infection-group they fall into. At 24 months, for those fulfilling the PCC Delphi research definition, SDQ and Chalder fatigue scores were higher, indicating more difficulties and SWEMBS scores were lower indicating poorer well-being. Self-rated health was poorer, and symptom severity and symptom impact were higher in those fulfilling the PCC Delphi research definition, irrespective of baseline PCR-test result. Therefore, CYP experiencing symptoms at 24 months were more likely to have poorer concurrent physical and mental health compared to those reporting no symptoms, irrespective of SARS-CoV-2 infection history. Importantly, too, the median global ratings of health, symptom impact and severity identified little difference by infection or vaccination status.

We found about one-quarter of CYP fulfilled the published PCC research definition at 24 months from the NP, PN, and NN groups, while 30% of those who were infected at least twice meet the PCC definition 24 months post-initial infection. This perhaps reflects that by 24 months a substantial proportion of CYP are likely to have been infected (at least once), even if some are asymptomatic and unaware that they have been infected[10]. It also speaks to the breadth of the PCC Delphi research definition which, in contrast to the WHO definition, does not require symptoms to have arisen within the first three months of infection[3]. The types of symptoms reported are known to be prevalent in the general adolescent population, raising questions about the role of the initial SARS-CoV-2 infection in their aetiology[22,23]. For example, those who did not fulfil the PCC Delphi research definition had SDQ scores broadly in-line with normative data from 11-to-15-year-old British CYP pre-pandemic (i.e. mean (SD): hyperactivity =

**Table 3 | Baseline characteristics by SARS-CoV-2 status, stratified by PCC[a] at 24 months (N (%), Mean (SD))**

| At 24 months | SARS-CoV-2 initial-negatives (NN; N = 4534) | | SARS-CoV-2 initial-positives (PN; N = 5177) | | SARS-CoV-2 positive & reinfected (PP; N = 519[b]) | | SARS-CoV-2 negative & infected (NP; N = 2402[b]) | |
|---|---|---|---|---|---|---|---|---|
| | Not meeting PCC definition (n = 3490) | Meeting PCC definition[a] (n = 1044) | Not meeting PCC definition (n = 3893) | Meeting PCC definition[a] (n = 1284) | Not meeting PCC definition (n = 363) | Meeting PCC definition[a] (n = 156) | Not meeting PCC definition (n = 1778) | Meeting PCC definition[a] (n = 624) |
| Prevalence (%) | | 23.0 | | 24.8 | | 30.1 | | 26.0 |
| Baseline characteristics: N (%) | | | | | | | | |
| *Sex* | | | | | | | | |
| Male | 1326 (85.0) | 234 (15.0) | 1548 (84.1) | 292 (15.9) | 145 (80.6) | 35 (19.4) | 676 (82.2) | 146 (17.8) |
| Female | 2164 (72.8) | 810 (27.2) | 2345 (70.3) | 992 (29.7) | 218 (64.3) | 121 (35.7) | 1102 (69.8) | 478 (30.3) |
| *Age at index-test (years)* | | | | | | | | |
| 11–14 | 1528 (80.8) | 363 (19.2) | 1802 (77.0) | 540 (23.1) | 194 (74.3) | 67 (25.7) | 991 (78.3) | 275 (21.7) |
| 15–17 | 1962 (74.2) | 681 (25.8) | 2091 (73.8) | 744 (26.2) | 169 (65.5) | 89 (34.5) | 787 (69.3) | 349 (30.7) |
| *Ethnicity* | | | | | | | | |
| White | 2,548 (76.5) | 785 (23.6) | 2903 (75.3) | 953 (24.7) | 284 (71.2) | 115 (28.8) | 1,451 (74.5) | 496 (25.5) |
| Asian/Asian British | 569 (79.0) | 151 (21.0) | 598 (75.8) | 191 (24.2) | 50 (65.8) | 26 (34.2) | 174 (73.1) | 64 (26.9) |
| Mixed | 184 (76.4) | 57 (23.7) | 204 (77.6) | 59 (22.4) | 16 (57.1) | 12 (42.9) | 88 (68.8) | 40 (31.3) |
| Black/African/ Caribbean | 123 (77.9) | 35 (22.2) | 109 (72.2) | 42 (27.8) | 7 (77.8) | 2 (22.2) | 42 (73.7) | 15 (26.3) |
| Other | 43 (81.1) | 10 (18.9) | 56 (64.4) | 31 (35.6) | 1 (50.0) | 1 (50.0) | 20 (74.1) | 7 (25.9) |
| Prefer not to say | 23 (79.3) | 6 (20.7) | 23 (74.2) | 8 (25.8) | 5 (100.0) | 0 (0.0) | 3 (60.0) | 2 (40.0) |
| *Region* | | | | | | | | |
| East Midlands | 237 (75.2) | 78 (24.8) | 288 (72.7) | 108 (27.3) | 30 (71.4) | 12 (28.6) | 131 (74.0) | 46 (26.0) |
| East of England | 758 (76.6) | 231 (23.4) | 663 (75.5) | 215 (24.5) | 53 (72.6) | 20 (27.4) | 473 (77.7) | 136 (22.3) |
| London | 766 (79.7) | 195 (20.3) | 743 (75.2) | 245 (24.8) | 60 (64.5) | 33 (35.5) | 302 (72.4) | 115 (27.6) |
| North East | 124 (77.5) | 36 (22.5) | 166 (79.1) | 44 (21.0) | 14 (77.8) | 4 (22.2) | 46 (68.7) | 21 (31.3) |
| North West | 385 (77.8) | 110 (22.2) | 458 (76.3) | 142 (23.7) | 45 (67.2) | 22 (32.8) | 171 (77.7) | 49 (22.3) |
| South East | 578 (78.6) | 157 (21.4) | 666 (75.4) | 217 (24.6) | 59 (71.1) | 24 (28.9) | 304 (71.2) | 123 (28.8) |
| South West | 141 (73.4) | 51 (26.6) | 220 (77.5) | 64 (22.5) | 28 (63.6) | 16 (36.4) | 93 (73.2) | 34 (26.8) |
| West Midlands | 302 (74.9) | 101 (25.1) | 373 (71.9) | 146 (28.1) | 38 (73.5) | 12 (24.0) | 141 (70.9) | 58 (29.2) |
| Yorkshire and the Humber | 199 (70.1) | 85 (29.9) | 316 (75.5) | 103 (24.5) | 36 (73.5) | 13 (26.5) | 117 (73.6) | 42 (26.4) |
| *IMD quintile* | | | | | | | | |
| 1 (most deprived) | 540 (71.2) | 218 (28.8) | 585 (70.9) | 240 (29.1) | 74 (68.5) | 34 (31.5) | 219 (65.4) | 116 (34.6) |
| 2 | 623 (76.4) | 192 (25.6) | 670 (72.2) | 258 (27.8) | 61 (67.0) | 30 (33.0) | 282 (73.6) | 101 (26.4) |
| 3 | 657 (76.5) | 202 (23.5) | 688 (73.1) | 253 (26.9) | 58 (59.2) | 40 (40.8) | 333 (72.4) | 127 (27.6) |
| 4 | 780 (80.8) | 185 (19.2) | 836 (75.5) | 272 (24.6) | 80 (76.2) | 25 (23.8) | 404 (76.8) | 122 (23.2) |
| 5 (least deprived) | 890 (78.3) | 237 (21.7) | 1114 (81.0) | 261 (19) | 90 (76.9) | 27 (23.1) | 540 (77.4) | 158 (22.6) |

[a]Using data from the questionnaire on symptoms and the EQ-5D-Y scale at the time of the questionnaire (i.e., approximately 24 months after the index PCR-test), PCC was operationalized as having at least 1 symptom and experiencing some/a lot of problems with respect to mobility, self-care, doing usual activities or having pain/discomfort or feeling very worried/sad. For the NN group we excluded the need for a positive PCR test.
[b]For the PP and NP groups, the timing of the (re)infection is uncertain.

3.8(2.2); emotional symptoms = 2.8 (2.1); peer problems = 1.5(1.4); conduct problems = 2.2(1.7))[24]. Likewise, the SWEMWBS score among those who did not fulfil the PCC Delphi research definition was similar to those reported pre-pandemic in 10-to-16-year-old Danish school children (mean = 23.33 (SD = 3.78))[25]. In previous studies which have included control groups, using either children with no serologic evidence of previous SARS-CoV-2 infection or children that never received a positive molecular diagnosis of SARS-CoV-2, persisting symptoms were reported by many CYP, irrespective of SARS-CoV-2 infection or antibody status[9,26–29]. Post-viral fatigue has also been described after the 1918 influenza pandemic, 'swine flu' epidemic of influenza A (H1N1), Ebolavirus, and tick-borne encephalitis. After mononucleosis, Q-fever, and giardiasis, prospective cohort studies report that 10–15% of patients experience moderate to severe disability, meeting the diagnostic criteria for post-infective fatigue syndrome[30]. For SARS-CoV-2, adult studies have found a correlation between autoantibodies and acute SARS-CoV-2 infection severity[31–33] and between viral load and PCC prevalence[34].

Despite the growing volume of research on lasting symptoms after COVID-19, a definition of PCC has not been universally agreed. In addition to the Delphi Consensus Definition for PCC in CYP[3], the UK National Institute for Health and Care Excellence[35], the World Health Organization[2,36], and the US Centers for Disease Control and Prevention[37]

**Table 4 | Validated scales 24 months post index-test by SARS-CoV-2 status, stratified by PCC[a] at 24 months (N (%), mean (SD))**

| At 24 months | SARS-CoV-2 initial-negatives (NN; N = 4534) | | SARS-CoV-2 initial-positives (PN; N = 5177) | | SARS-CoV-2 positive & reinfected (PP; N = 519[e]) | | SARS-CoV-2 negative & infected (NP; N = 2402[e]) | |
|---|---|---|---|---|---|---|---|---|
| | Not meeting PCC definition (n = 3490) | Meeting PCC definition[a] (n = 1044) | Not meeting PCC definition (n = 3893) | Meeting PCC definition[a] (n = 1284) | Not meeting PCC definition (n = 363) | Meeting PCC definition[a] (n = 156) | Not meeting PCC definition (n = 1778) | Meeting PCC definition[a] (n = 624) |
| *Characteristics at 24-months post index-test: Mean (SD) or Median (IQR)* | | | | | | | | |
| **SDQ** | | | | | | | | |
| Total difficulties | 10.9 (5.9) | 17.1 (6.1) | 10.5 (5.8) | 16.4 (6.0) | 10.4 (6.0) | 17.5 (6.4) | 10.8 (5.9) | 16.7 (6.4) |
| Emotional symptoms | 3.4 (2.4) | 5.9 (2.5) | 3.3 (2.4) | 5.7 (2.4) | 3.3 (2.4) | 6.2 (2.4) | 3.3 (2.3) | 5.9 (2.6) |
| Conduct problems | 1.5 (1.4) | 2.2 (1.8) | 1.4 (1.5) | 2.2 (1.7) | 1.5 (1.5) | 2.6 (1.9) | 1.5 (1.5) | 2.2 (1.8) |
| Hyperactivity/inattention | 3.9 (2.5) | 5.7 (2.5) | 3.8 (2.5) | 5.5 (2.5) | 3.7 (2.6) | 5.6 (2.5) | 3.9 (2.6) | 5.6 (2.6) |
| Peer relationship problem | 2.2 (1.8) | 3.3 (2.2) | 2.1 (1.7) | 3.0 (2.0) | 2.0 (1.8) | 3.2 (2.2) | 2.0 (1.8) | 3.0 (2.0) |
| SWEMWBS | 21.8 (4.0) | 18.8 (3.4) | 21.9 (3.9) | 19.1 (3.5) | 22.3 (4.4) | 18.6 (3.1) | 21.9 (4.1) | 19.0 (3.5) |
| Chalder fatigue scale | 12.7 (4.4) | 18.1 (5.6) | 12.8 (4.3) | 18.5 (5.7) | 12.6 (4.3) | 19.7 (5.7) | 12.7 (4.2) | 18.6 (5.8) |
| Self-rated health[b] | 85 (80, 95) | 70 (50, 80) | 90 (80, 95) | 70 (55, 80) | 90 (80, 95) | 70 (55, 80) | 90 (80, 95) | 70 (55, 80) |
| Symptom severity[b,c] | 50 (30,60) | 60 (40,70) | 50 (30, 60) | 60 (40, 70) | 40 (20, 60) | 60 (45, 70) | 50 (30, 60) | 60 (40, 70) |
| Symptom impact[b,d] | 40 (10,60) | 50 (30,80) | 40 (10, 60) | 50 (30, 80) | 30 (10, 60) | 60 (30, 80) | 40 (20, 60) | 60 (40, 80) |

A higher SDQ score indicates more problems; a higher SWEMWBS score indicates better mental well-being; a higher fatigue score is more severe.

SDQ strengths and difficulties questionnaire, SWEMWBS short Warwick–Edinburgh mental wellbeing scale.

[a]Using data from the questionnaire on symptoms and the EQ-5D-Y scale at the time of the questionnaire (i.e., approximately 24 months after the index PCR test), PCC was operationalized as having at least 1 symptom and experiencing some/a lot of problems with respect to mobility, self-care, doing usual activities or having pain/discomfort or feeling very worried/sad. For the NN group, we excluded the need for a positive PCR test. Number of symptoms and the EQ-5D-Y scale at 24 months are not shown in this table, as they are part of the definition of PCC.

[b]Reported as median(IQR), scored on a scale of 0 (worst) to 100 (best) for self-rated health; 0 (not severe at all) to 100 (extremely severe) for symptom severity; 0 (no impact) to 100 (extreme impact) for symptom impact.

[c]For the PP and NP groups, the timing of the (re)infection is uncertain.

[d]Based on sub-sample N = 4977.

[e]Based on sub-sample N = 4972.

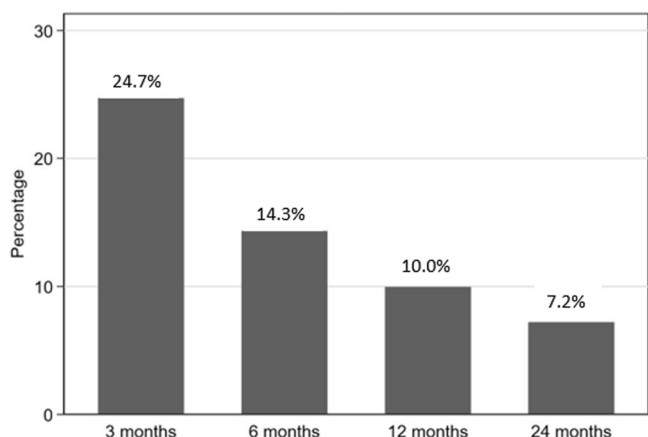

**Fig. 3 | Proportion of index-test positive CYP fulfilling the PCC definition at 3, 6, 12, and 24 months (N = 943)[a].** [a]At each time point, we show the proportion of CYP that fulfil the PCC definition at the time point in question and all prior time points. For example, at 6 months 14.3% of CYP (i.e. 135/943) fulfilled the PCC definition at both 3 and 6 months; at 12 months this was 10.0% (94/943); at 24 months it was 7.2% (68/943). Sample (N = 943) consists of index-test positive CYP with data at 3, 6, 12, and 24 months post index-test.

have published operational clinical definitions of post-COVID-19 condition, but all three are different and only the WHO definition applies to CYP. Of 295 studies reviewed in November 2022, almost two-thirds did not comply with any of the definitions[4], highlighting a problem in comparing outcomes between studies of PCC due to differences in definition. Judging by a more recent review from April 2023[38], the Delphi consensus research definition is the only one currently being used for CYP. An important difference between the WHO and Delphi consensus research definition is that the WHO definition[2,36] requires symptoms of PCC to have arisen within three months of the initial infection. This means that new symptoms arising 6 or 12 months post-infection should not be considered as PCC. In our data, including participants that completed questionnaires at every possible time point from 3 months post-index test, only 7% of CYP fulfilled the definition of PCC at 3, 6, 12, and 24 months.

It has been argued that, partly due to a lack of testing, it is likely that current estimates are underestimating the true burden of those with PCC[1]. Moreover, cross-sectional studies provide only a static prevalence of symptoms of PCC, and as these are likely to vary over time, such studies may underestimate the true burden of PCC within populations. That is why we have published CLoCk data as both 'snap shots' of the whole study cohort and longitudinal reports on smaller sub-sets with data at multiple timepoints[8]. These latter analyses show that in CYP, the prevalence of adverse symptoms reported at the time of a positive PCR-test declined over the first 12 months, but some test-positives and test-negatives reported adverse symptoms for the first time at six- and 12 months post-test, particularly tiredness, shortness of breath, poor quality of life, poor well-being and fatigue.

Taking together all available data so far, our findings continue to raise important questions about the role of SARS-CoV-2 infection and ongoing symptoms reported by CYP as well as definitions of PCC in CYP. That some CYP will go on the develop persistent symptoms after acute SARS-CoV-2 infection is undeniable, consistent with other post-viral syndromes, and there is emerging evidence of viral persistence in children and immunological biomarkers[8] in adults[8]. It is, however, unlikely that SARS-CoV-2 infection is responsible for all reported symptoms in 54-65% of CYP in our cohort. Whilst some symptoms, such as loss of taste or smell, are strongly associated with SARS-CoV-2 infection, these improve with time, while most of the other reported symptoms are non-specific and often commonly reported in adolescents, even before the current pandemic. Importantly, too, our findings clearly demonstrate no impact on self-rated health, symptom severity or symptom impact in CYP when assessed by infection or

vaccination status[8]. Nonetheless, CYP reporting symptoms at 24 months follow-up, irrespective of their SARS-CoV-2 infection status, were more likely to have poorer self-rated concurrent physical and mental health compared to those who did not report symptoms at 24 months, highlighting a wider need for assessment and interventions to support the physical and mental health of CYP in the community during critical developmental periods, irrespective of SARS-CoV-2 infection or pandemic effects.

With a response rate of 40.7% we have been able to retain CYP in our study over a long period. Moreover, previous analyses indicate that findings from the CLoCk study are broadly generalisable to CYP in England[39]. Nonetheless, we have stated our study limitations previously, including those of potential selection bias and attrition over time[8]. In this study, we additionally acknowledge issues around misclassification. For example, some CYP categorised as never testing positive for SARS-CoV-2 infection may not be true 'negatives' (i.e., NN). Thus, reported differences between the NN group and the infected groups (PP, PN, and NP) are conservative and might be larger if we had a confirmed group of true 'negatives'. False-negative serologic responses are possible after initial infection, negative molecular tests can be false-negative, and lack of universal testing and lack of specific symptoms as well as asymptomatic infection may all lead to misclassification[9,40]. Similarly, infections may have gone undetected, especially after the cessation of free community testing in England from April 2022. Therefore, misclassification into the four infection status groups may have occurred, especially in the NN group. Our study design exploits the SARS-CoV-2 testing dataset held by UKHSA and made for a cost-efficient study. However, this also comes with its own unique limitations and unfortunately, we had no control and limited data on why, where, when, and how often CYP tested for SARS-CoV-2. Therefore, for example, unknown to us, the reason for index- and subsequent testing could differ among our infection-status groups. Additionally, as ~45% of the analytical sample enrolled in CLoCk 12 months post index-PCR test, we actively chose not to focus on comparisons using retrospectively reported data (e.g., symptoms at the time of testing and pre-testing conditions). We acknowledge that some of our estimated prevalence of PCC in the NP and PP groups could be due to acute infection because we are uncertain of the timing of the (re)infection. We also did not examine menstruation, and some symptoms may be attributable to pre-menstrual syndrome given the high proportion of girls. Importantly, the study primarily focuses on CYP in England, and the findings may not be directly applicable to other populations or countries with different healthcare systems, vaccination rates, and demographics. Finally, the study primarily relied on self-reported symptoms and questionnaires, which may lack the precision of detailed, objective clinical assessments. However, it could be argued that this a strength of the study at this point given that there is no definitive diagnostic test to determine if a patient has PCC and we need to learn about new emerging symptoms, or re-emerging, chronic symptoms from those who are experiencing them.

In conclusion, we found that around 25–30% of CYP fulfilled the Delphi research PCC definition at 24 months, but only 7.2% met the definition at all time points and those young people had multiple symptoms. However, there was no evidence that self-rated health, symptom severity, or symptom impact varied by infection or vaccination status. Our study highlights the relatively high prevalence of non-specific symptoms in our cohort of CYP. Further studies are needed to understand the pathophysiology, develop diagnostic tests, and identify effective interventions for PCC in CYP.

## Data availability
An anonymised version of the Children & Young People with Long COVID-19 (CLoCk) data will shortly be available via the UK Data Service (study number: 9203; https://doi.org/10.5255/UKDA-SN-9203-1).

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

## Acknowledgements

Michael Lattimore, UKHSA, as Project Officer for the CLoCk study. Olivia Swann designed the elements of the ISARIC Paediatric COVID-19 follow-up questionnaire, which were incorporated into the online questionnaire used in this study to which all the CLoCk Consortium members contributed. This

work is independent research jointly funded by the National Institute for Health and Care Research (NIHR) and UK Research & Innovation (UKRI), who have awarded funding grant number COVLT0022. All research at Great Ormond Street Hospital NHS Foundation Trust and UCL Great Ormond Street Institute of Child Health is made possible by the NIHR Great Ormond Street Hospital Biomedical Research Centre. For the purpose of open access, the author has applied a Creative Commons Attribution (CC BY) licence to any Author Accepted Manuscript version arising. S.M.P.P. is supported by a UK Medical Research Council Career Development Award (ref: MR/P020372/1) and a Senior Non-clinical fellowship (ref: MR/Y009398/1). The views expressed in this publication are those of the authors and not necessarily those of the NHS, the NIHR, UKRI, or the Department of Health and Social Care.

## Author contributions

Conceptualisation: T.Ste., S.M.P.P., M.D.N., E.D., A.H., E.W., I.H., T.F., T.S., and T.C. CLoCk Consortium, R.S.; Methodology: S.M.P.P., M.D.N., R.S.; Software: S.M.P.P., M.D.N.; Validation: S.M.P.P., M.D.N.; Formal analysis: S.M.P.P., M.D.N.; Investigation: T.Ste., R.S.; Resources: L.X. Data curation: S.M.P.P., M.D.N.; Writing (draft): T.Ste., S.M.P.P., M.D.N., R.S.; Writing (review): T.Ste., S.M.P.P., M.D.N., E.D., A.H., E.W., I.H., T.F., T.S., T.C., L.X., L.F.S., R.S.; Supervision: T.Ste., S.M.P.P., R.S. All authors had full access to the data in the study and had the final responsibility for the decision to submit for publication. S.M.P.P. and M.D.N. accessed and verified the data.

## Competing interests

Terence Stephenson is Chair of the Health Research Authority and, therefore, recused himself from the Research Ethics Application. Trudie Chalder has been a member of the National Institute for Health and Care Excellence Committee for long COVID-19. She has written self-help books on chronic fatigue and has done workshops on chronic fatigue and post-infectious syndromes. All remaining authors have no competing interests.

## Additional information

[1]UCL Great Ormond Street Institute of Child Health, London, UK. [2]Division of Surgery & Interventional Science, Faculty of Medical Sciences, University College London, London, UK. [3]Nuffield Department of Primary Care Health Sciences, University of Oxford, Oxford, UK. [4]Department of Infectious Diseases, Imperial College London, London, UK. [5]Department of Psychiatry, University of Cambridge, Hershel Smith Building Cambridge Biomedical Campus, Cambridge, UK. [6]University College London Hospitals NHS Foundation Trust, London, UK. [7]Department of Psychological Medicine, Institute of Psychiatry, Psychology and Neuroscience, King's College London, De'Crespigny Park, London, UK. [8]Immunisations and Vaccine Preventable Diseases, UK Health Security Agency, 61 Colindale Avenue, London, UK. [15]These authors contributed equally: Terence Stephenson, Snehal M. Pinto Pereira. *A list of authors and their affiliations appears at the end of the paper.
✉ e-mail: snehal.pereira@ucl.ac.uk

## CLoCk Consortium

**Marta Buszewicz[1], Esther Crawley[9], Bianca De Stavola[1], Shruti Garg[10], Dougal Hargreaves[11], Michael Levin[11], Vanessa Poustie[12], Malcolm Semple[12], Kishan Sharma[13,16] & Olivia Swann[14]**

[9]University of Bristol, Bristol, UK. [10]University of Manchester, Manchester, UK. [11]Imperial College London, London, UK. [12]University of Liverpool, London, UK. [13]Manchester University NHS Foundation Trust, Manchester, UK. [14]University of Edinburgh, Edinburgh, UK. [16]Deceased: Kishan Sharma.

