## [Peer Review File · Communications Medicine]

REVIEWERS' COMMENTS:

Reviewer #1:

Remarks to the Author:

Overall comments

This is a well written analysis comparing long COVID symptom type, number, and impact across a cohort of children and young people (CYP) followed for 24 months and divided into four groups of based on serial SARS-CoV-2 testing results from United Kingdom Health Security Agency data and self-report: 'initial test-negatives with no subsequent positive test' (NN); 'initial test-negatives with a subsequent positive test' (NP); 'initial test-positives with no report of subsequent re-infection' (PN); and 'initial test-positives with report of subsequent re-infection' (PP). CYP were also classified as meeting or not meeting the Delphi consensus post-COVID condition (PCC) definition across the four groups. The Delphi criterion "a history of probable or confirmed SARS-CoV-2 infection" was waived for the NN group with the assumption that given current population seroprevalence rates, some individuals in the NN group likely had infection prior to study enrollment or had undetected infection during enrollment.

The major findings of this analysis are

- 1) Members of all four groups had some symptoms at 24 months (20-25% of all four groups reported 3+ symptoms at 24 months). 7.2% of CYP with a positive test (NP, PN, PP) fulfilled PCC Delphi definition at 3-, 6-, 12, and 24 months. 23% of NN group and 25% of PN met Delphi PCC definition at 24 months. CYP meeting the Delphi PCC definition at 24 months had increased symptom impact, lower quality of life, and increased fatigue compared to CYP not meeting Delphi PCC at 24 months.
- 2) At 24 months, tiredness, trouble sleeping, shortness of breath, and headaches were the most commonly reported symptoms. Symptom profiles differed by infection group.
- 3) PCC was more common in older than younger CYP, more common in the most vs least deprived quintile, and twice as common in females compared to males. No significant patterns identified in ethnicity.
- (4) The PP group reported more symptoms compared to the NP, PN, and NN groups.
- (5) Irrespective of infection status, vaccinated versus unvaccinated, there were no obvious trends in numbers of symptoms reported, health, well-being or symptom impact or severity at 24-months. Chronology of vaccine administration compared to infection could not be confirmed and vaccination appears to have been defined as receipt of any dose.

PCC in children is a relatively new and difficult area of study. These findings provide valuable insight into the evolution of PCC symptoms and duration in children over time. They also highlight the prevalence of PCC type symptoms in general, even in the group with negative testing (although prior or undetected SARS-CoV-2 infection in this group is likely). This analysis shows the variety of symptoms, symptom

combinations, and impact of symptoms on function. These are all important data for health care providers and public health practitioners.

Specific comments:

A draft with line numbers was uploaded by this reviewer to assist with specific feedback:

Abstract

Background

I would introduce the concept of Post-COVID conditions here as you do not mention the term PCC until findings

Methods:

Line 46: would spell out CloCK at first use

Would add here that you use the Delphi PCC definition (and waved the "a history of probable or confirmed SARS-CoV-2 infection" for the NN group if room).

Can you add something on how impact was assessed/ mention that health scales were performed? Or at least mention that symptom severity was assessed?

Findings

Can you add some findings regarding severity of symptoms?

Introduction

Lines 116-129: A lot of this seems like methods. The seroprevalence discussion seems appropriate for intro and does explain the NN group. But for the intro/hypotheses could you leave the group description vague and say something like you compared symptoms across groups categorized by infection status including a test negative group that was presumed seropositive. Or something like that.

Line 117: From the reference it appears that this seroprevalence does not refer to the CloCK study cohort itself correct? If so could you add something to clarify like, "based on a UK population-based survey" or something like that to clarify the data is not coming from the CloCK cohort? Do you have seroprevalence data on the CloCK cohort, particularly the NN group, that could be included in results?

Methods

Line 143: can you provide data on reason for testing? For both initial tests and subsequent? Are these screening tests of asymptomatic children or tests during illness? Could the reason for testing differ among the 4 groups?

Line 155: Are these Lateral Flow Tests tests administered at home? Were survey respondents able to record all home tests done during study period?

Line 160: Did the questionnaire include information on underlying medical conditions? And information on severity of acute SARS-CoV-2 infections. Do you know if underlying conditions or SARS-CoV-2 infection severity differed among the 4 groups?

Results

Line 217: based on tables I assume COVID-19 vaccination was defined as receipt of any dose (would add that to methods somewhere)

Line 238: Perhaps I am missing something here but why does the test positive group only include PN and not NP and PP as well?

Discussion

Line 360: there appears to be a typo? "NO_PRINTED_FORM"

Line 425: It is interesting that the authors found no difference in symptom number or severity across any of the 4 groups by vaccination status even though adult studies and some pediatric studies have found that COVID-19 vaccination is associated with a reduced likelihood of PCC (Notarte KI, Catahay JA, Velasco JV, et al. Impact of COVID-19 vaccination on the risk of developing long-COVID and on existing long-COVID symptoms: A systematic review. *eClinicalMedicine*. 2022;53) and (Razzaghi H et al. Vaccine Effectiveness Against Long COVID in Children: A Report from the RECOVER EHR Cohort. *medRxiv* [Preprint]. 2023 Sep 28:2023.09.27.23296100. doi: 10.1101/2023.09.27.23296100. Update in: *Pediatrics*. 2024 Jan 16;: PMID: 37808803; PMCID: PMC10557822.) This analysis could not confirm if COVID-19 vaccination occurred before or after SARS-CoV-2 infections or subsequent infections. Vaccine status was defined as receipt of any dose and compared across all 4 groups. It does appear that the NN group had the highest percentage of participants with 3+ vaccine doses while the NP group had the lowest although not noted if these are significant difference. I wonder if there would be differences seen in symptom prevalence at 24 months when comparing the positive testing groups to the NN group as a referent control group and defining vaccination as receipt of at least 2 doses.

Anna Yousaf, MD

Severe Illness and Multisystem Inflammatory Syndrome (SIM) Team

Epidemiology Branch

Coronavirus and Other Respiratory Viruses Division

National Center for Immunization and Respiratory Diseases

Centers for Disease Control and Prevention

pgy6@cdc.gov

Reviewer #2:

Remarks to the Author:

I think this is an important paper to get out because of the limited amount of information on PCC in CYP and the size of this study to get a broader understanding of the problem even two years after infection. The results are generally consistent with other studies in adults, but help to add key missing information for this age group. That being said, I do think the methods and results sections need several clarifications. Because this is a follow-up paper to a larger study that has already been published, it is expected that not all of the very detailed methods would be included. However, there are some details that could be added in the context of this particular study that would be extremely helpful to get a better understanding of the methods and results.

Introduction

1. Line 118 – It is possible to differentiate between antibodies from infection and vaccination status with specific testing, but was not possible within this study. This should be clarified.

<https://www.cdc.gov/coronavirus/2019-ncov/hcp/testing/antibody-tests-guidelines.html>

Methods & Results

2. As a reader it was difficult to tell how many children just received the 24mo survey, how many were followed longitudinally starting at 3mo and how many started somewhere in between. You can piece it together between the footnote in Figure 1 and one reference to a subset of participants followed over a longer period of time (lines 245-253), but it should be much clearer in the methods or could be added to Figure 1.

3. The main concern is recall bias. If there is differential methods between the groups, for instance far more NN's completed only 24mo whereas more of the PP's completed multiple surveys from baseline to 24mo, there may be differences in the way children are answering the 24mo survey. Lines 148-150 in the Methods attempt to explain it, but it would help to clarify the inclusion criteria for how long CYP had to respond to the 'retrospective baseline survey' and in the results what that time actually was. Data should be presented on the range and mean/median time of from infection to 'baseline', 'baseline' to '24mo' survey and from 24mo survey to previous survey (if not baseline) for each group or examined and noted if no differences are present.

4. Line 188 – What does 'examining the data visually' actually mean (ie did you investigate further only the ones that looked obviously different)?

5. The NP and PP groups were not used in presenting many of the PCC results because, the date of the infection after baseline was not asked so investigators could not guarantee that the symptoms reported

at 24mo were due to PCC or an infection happening in the last month (line 193-196). Was any sort of sensitivity analyses conducted for CYP that had dates of their PCR tests recorded in the database (as opposed to those you only reported a home lateral test), to see how the results would have changed if these cases had been included? It was a huge loss of data to not include these groups and presumably, the vast majority of those reporting an infection after baseline were not infected/reinfected in the 3 months prior to taking their 24mo survey.

Discussion & General Comments/Edits

6. Data related to vaccination status is not consistent with multiple studies in adults. Do the authors think this is due to limitations in the study design (unable to determine temporality due to self-reported test dates (line 200), biological reasons, or both? It also appears from the data that the vaccination rates are much higher in the NN group than the PP group (lines 217-218) which also report the lowest and highest levels of PCC correspondingly (there also seems to be differences in Sup Table 2 on number of symptoms and number of vaccine doses). Was this considered in any sort of multi-variable modeling? Can data be presented on vaccination status at initial baseline testing? This could at least establish who was vaccinated/unvaccinated at initial infection and be able to compare PCC status at 24mo. It also seems possible to examine temporality amongst the subset of participants with complete follow-up.

7. There seems to be a slight disconnect for group NN throughout the paper. While they are presumably the 'negatives' the assumption throughout the paper (not just what was noted in the discussion) seems to be that the NN group represents children who just never tested positive vs. those who have truly never been infected. While this is certainly likely, it may be good to set this up more in the introduction.

8. Line 301 – compared, not compare

Tables and Figures

9. Table 1 – Were there any significant differences in responders vs. non-responders? If so, they should be noted.

10. Table 3 – The NN and PP groups look almost identical for all the standardized scoring (and also close on the prevalence of PCC). Since these standardized surveys were in use before the pandemic, did the group ever compare the NN groups scores to any available for the same age group before the pandemic? It might give an indication of what may be due to undocumented infection/stress of being a kid during a pandemic and what is just general CYP stress or ailments.

Reviewer #1

Overall comments:

Response: Thank you for your detailed and fair review of your manuscript. Below we explain how we have improved the manuscript based on comments provided.

Specific comments:

Abstract

Background

I would introduce the concept of Post-COVID conditions here as you do not mention the term PCC until findings

Response: We have edited the first line to state (page 3 lines 38-40): Most children and young people (CYP) in the United Kingdom have been infected with SARS-CoV-2 and some continue to experience impairing symptoms after infection (often described as Post-COVID-19 condition (PCC)).

Methods:

Line 46: would spell out CloCK at first use

Response: This has been done.

Would add here that you use the Delphi PCC definition (and waived the "a history of probable or confirmed SARS-CoV-2 infection" for the NN group if room).

Response: We have edited the text (on page 3 lines 50-52) to now state: "PCC was determined by operationalising the PCC Delphi research definition in CYP (waiving the need for a history of SARS-CoV-2 infection for the NN group)."

Can you add something on how impact was assessed/ mention that health scales were performed? Or at least mention that symptom severity was assessed?

Response: We have edited the text (on page 3 lines 52-54) to now state: "Symptom severity and impact were assessed via visual analogue scales; validated health scales (e.g., Strengths and Difficulties Questionnaire and Chalder Fatigue Scale) were also recorded."

Findings

Can you add some findings regarding severity of symptoms?

Response: We have edited the text (on page 4 lines 65-66) to now state: "Symptom severity and impact were higher in those fulfilling the PCC definition, irrespective of infection history."

Introduction

Lines 116-129: A lot of this seems like methods. The seroprevalence discussion seems appropriate for intro and does explain the NN group. But for the intro/hypotheses could you leave the group description vague and say something like you compared symptoms across groups categorized by infection status including a test negative group that was presumed seropositive. Or something like that.

Response: We have edited this paragraph (page 7 lines 117-132) to now read: "In previous CLoCk publications, test-positive CYP were compared to laboratory-confirmed SARS-CoV-2 test-negative CYP. However, according to the COVID-19 Schools Infection Survey in England, by June 2022, 82-99% of CYP had SARS-CoV-2 antibodies, consistent with prior infection and/or COVID-19 vaccination.¹⁰ Therefore, it is now no longer possible to compare test-positive with test-negative CYP from the CLoCk study. Instead, here we report on symptoms self-reported 24-months post-testing on over 12,600 CYP categorized into four groups by infection status over the 24-month period, including a group that never tested positive. By operationalising the Delphi definition of PCC 24-months post PCR-testing, we are able to address questions about the profile, persistence and impact of symptoms post-infection."

Line 117: From the reference it appears that this seroprevalence does not refer to the CloCk study cohort itself correct? If so could you add something to clarify like, "based on a UK population-based survey" or something like that to clarify the data is not coming from the CloCk cohort? Do you have seroprevalence data on the CloCk cohort, particularly the NN group, that could be included in results?

Response: We have edited the text (on page 7 lines 118-120) to now read "However, according to the COVID-19 Schools Infection Survey in England, reporting in June 2022 on data from March 2022, 99% of teenagers had SARS-CoV-2 antibodies, consistent with prior infection and/or COVID-19 vaccination.¹⁰", thus clarifying that this statement is not based on data from the CLoCk study. We have no seroprevalence data on the CLoCk cohort as they were recruited and responded to questionnaires online and were never seen in person. We acknowledge this limitation in the discussion (on page 19 lines 430-432): "However, ... and unfortunately, we had no control and limited data on why, when and how often CYP tested for SARS-CoV-2."

Methods:

Line 143: can you provide data on reason for testing? For both initial tests and subsequent? Are these screening tests of asymptomatic children or tests during illness? Could the reason for testing differ among the 4 groups?

Response: Thank you for this comment. Unfortunately, we do not have reliable information on reason for testing. However, the reviewer does raise a valid point which we now acknowledge in the discussion (on page 19 lines 429-434): "Our study design exploits the SARS-CoV-2 testing dataset held by UKHSA and made for a cost-efficient study. However, this also comes with its own unique limitations and unfortunately, we had no control and limited data on why, where, when and how often CYP tested for SARS-CoV-2. Therefore, for example, unknown to us, the reason for index- and subsequent testing could differ among our infection-status groups."

Line 155: Are these Lateral Flow Tests tests administered at home? Were survey respondents able to record all home tests done during study period?

Response: We did not ask CYP where Lateral Flow Tests were administered, however we did ask “Have you had a positive COVID-19 test result?” allowing the CYP to self-report on their Lateral Flow (&/or PCR) tests. We clarify in the methods (on page 8 lines 159-163): “The dataset held by UKHSA can track repeated PCR testing longitudinally within individuals. In addition, at each contact, we asked CYP if they ever tested positive for SARS-CoV-2 (thereby allowing CYP to report on any PCR or Lateral Flow Test taken). Using information on subsequent testing held by UKHSA and from self-report, we divided the CYP into four groups...” We also acknowledge in the discussion (on page 19 lines 431-432): “we had no control and limited data on why, where, when and how often CYP tested for SARS-CoV-2.”

Line 160: Did the questionnaire include information on underlying medical conditions? And information on severity of acute SARS-CoV-2 infections. Do you know if underlying conditions or SARS-CoV-2 infection severity differed among the 4 groups?

Response: Thank for you this question on which we have reflected on at length. We do have some information on underlying medical conditions and severity of acute SARS-CoV-2 infections. However, a substantial proportion of our sample (~45%) would have reported on these things 12-months after their index-PCR test. Therefore, on balance, we felt that an analysis of underlying conditions or SARS-CoV-2 infection severity by the 4 infection groups would add more in terms of uncertainty due to recall bias, than providing more clarity to our manuscript. However, we acknowledge the point made by the reviewer and explain why this analysis was not undertaken (on page 19 lines 434-436): “Additionally, as ~45% of the analytical sample enrolled into CLoCk 12-months post index-PCR test, we actively chose not to focus on comparisons using retrospectively reported data (e.g., symptoms at time of testing and pre-testing conditions).”

Results:

Line 217: based on tables I assume COVID-19 vaccination was defined as receipt of any dose (would add that to methods somewhere)

Response: We had edited the methods (on page 10 lines 210-212) to state: “Finally, study participant characteristics on validated scales 24-months post index-test, were stratified by infection status groups and vaccination status (dosage: 0, 1, 2, 3+) at 24-months.”

Line 238: Perhaps I am missing something here but why does the test positive group only include PN and not NP and PP as well?

Response: Thank you for raising this. Based on this reviewer’s comment and those from reviewer 2 we have now edited the text and Table 3 to include the PP and NP groups (along with the NN and PN groups). The results (page 12 lines 251-253)) now state: “Approximately one-quarter of CYP fulfilled the broad Delphi research definition of PCC 24-months post index-testing in the NN, PN and NP groups (Table 3); while 30% of the PP group met this broad PCC definition 24-months after their first infection.”

Discussion

Line 360: there appears to be a typo? "NO_PRINTED_FORM"

Response: This has been corrected.

Line 425: It is interesting that the authors found no difference in symptom number or severity across any of the 4 groups by vaccination status even though adult studies and some pediatric studies have found that COVID-19 vaccination is associated with a reduced likelihood of PCC (Notarte KI, Catahay JA, Velasco JV, et al. Impact of COVID-19 vaccination on the risk of developing long-COVID and on existing long-COVID symptoms: A systematic review. *eClinicalMedicine*. 2022;53) and (Razzaghi H et al. Vaccine Effectiveness Against Long COVID in Children: A Report from the RECOVER EHR Cohort. *medRxiv [Preprint]*. 2023 Sep 28:2023.09.27.23296100. doi: 10.1101/2023.09.27.23296100. Update in: *Pediatrics*. 2024 Jan 16;: PMID: 37808803; PMCID: PMC10557822.) This analysis could not confirm if COVID-19 vaccination occurred before or after SARS-CoV-2 infections or subsequent infections. Vaccine status was defined as receipt of any dose and compared across all 4 groups. It does appear that the NN group had the highest percentage of participants with 3+ vaccine doses while the NP group had the lowest although not noted if these are significant difference. I wonder if there would be differences seen in symptom prevalence at 24 months when comparing the positive testing groups to the NN group as a referent control group and defining vaccination as receipt of at least 2 doses.

Response: Vaccination has been well established to prevent SARS-CoV-2 infection in children and adults. This is primary prevention. One would anticipate that if the vaccine stops an individual getting infected, then the individual would not subsequently develop long COVID. This is what both these studies showed.

The Notarte et al 2022 systematic review included only adults (>18 years) infected by SARS- CoV-2 and diagnosed with real-time reverse tran- scription-polymerase chain reaction (RT-PCR) assay. Individuals could have been hospitalised or not by SARS-CoV-2 acute infection. It concluded "Low level of evidence (grade III, case-controls, cohort studies) suggests that vaccination before SARS-CoV-2 infection could reduce the risk of subsequent long-COVID. The impact of vaccination in people with existing long-COVID symptoms is still controversial, with some data showing changes in symptoms and others did not."

*Razzaghi H et al. Vaccine Effectiveness Against Long COVID in Children: A Report from the RECOVER EHR Cohort. *Pediatrics*. March 2024 reported the following: for children with evidence of vaccination and no history of prior SARS-CoV-2 infection, Vaccine Effectiveness was 35% for the combined age groups with greater effect in adolescents than younger children. For children known to be completely vaccinated (ie, 2+ recorded doses), Vaccine Effectiveness was 45% against probable long COVID within 12 months.*

*The more difficult question relates to secondary prevention – does vaccination in individuals who are already suffering long COVID ameliorate their symptoms? We are aware of the following analysis in adults which suggests not only primary prevention but also that secondary mitigation is plausible: Ayoubkhani D, Bermingham C, Pouwels KB, et al. Trajectory of long covid symptoms after covid-19 vaccination: community based cohort study. *BMJ* 2022;377:e069676. For pre-existing long COVID in*

adults, vaccination was associated with a 12.8% decrease in self-reported prevalence of long COVID; a second dose was associated with an 8.8% decrease.

However, we are not aware of any publication relating to children which addresses this question. We would therefore have liked to address this but because we don't know when vaccination occurred in relation to infection/re-infection we are unable to have confidence in the suggested analysis.

Reviewer #2:

Overall Comments:

Response: Thank you for acknowledging that this is an important paper to get out due to its novelty (in terms of study size, age range examined and length of follow up time). Below you will find a point-by-point response to all comments made which we believe have clarified the methods and results sections.

1. Line 118 – It is possible to differentiate between antibodies from infection and vaccination status with specific testing, but was not possible within this study. This should be clarified.

<https://www.cdc.gov/coronavirus/2019-ncov/hcp/testing/antibody-tests-guidelines.html>

Response: Thank you for drawing our attention to this so that we can clarify this point. We have amended this section to read "In previous CLoCk publications, test-positive CYP were compared to laboratory-confirmed SARS-CoV-2 test-negative CYP. However, according to the COVID-19 Schools Infection Survey in England, reporting in June 2022 on data from March 2022, 99% of teenagers had SARS-CoV-2 antibodies, consistent with prior infection and/or COVID-19 vaccination.10 Of the 99.0% of secondary school pupils with SARS-CoV-2 antibodies, 65% had been vaccinated and 34% were unvaccinated. Vaccinated pupils could develop antibodies through vaccination and/or natural infection, whereas unvaccinated pupils will only have antibodies following natural infection. Although it is possible in principle to differentiate between the antibodies from infection and vaccination status with specific testing, it was not possible within this study."

2. As a reader it was difficult to tell how many children just received the 24mo survey, how many were followed longitudinally starting at 3mo and how many started somewhere in between. You can piece it together between the footnote in Figure 1 and one reference to a subset of participants followed over a longer period of time (lines 245-253), but it should be much clearer in the methods or could be added to Figure 1.

Response: Thank you for this comment. We have now made a new Figure (supplementary figure 1), that explains which groups of CYP started at 3- 6- and 12-months post index-testing.

3. The main concern is recall bias. If there is differential methods between the groups, for instance far more NN's completed only 24mo whereas more of the PP's completed multiple surveys from baseline to 24mo, there may be differences in the way children are answering the 24mo survey. Lines 148-150 in the Methods attempt to explain it, but it would help to clarify the inclusion criteria for how long CYP had to respond to the 'retrospective baseline survey' and in the results what that time actually was. Data should be presented on the range and mean/median time of from infection to

'baseline', 'baseline' to '24mo' survey and from 24mo survey to previous survey (if not baseline) for each group or examined and noted if no differences are present.

Response: With the addition of the new Supplementary Figure 1 and the edited text (on page 8 lines 150-153) that states: "After obtaining written informed consent, CYP completed an online questionnaire about their health at the time of their SARS-CoV-2 PCR test ("baseline"; retrospectively reported) and contemporaneously at approximately 3-, 6-, 12- and 24-months after their index-PCR test", we hope that it is now clear to the readers that all CYP completed the 24-month questionnaire contemporaneously. Additionally, in the results (page 11 lines 229-230) we state "24-month follow-up questionnaires were returned at a median of 104.1 weeks [IQR:103.0, 105.6] after the index-PCR test". Nonetheless, we agree that recall bias is a concern. For this reason, we were careful in the analyses we undertook, for example in the discussion (page 19 lines 434-436) we now state: "Additionally, as ~45% of the analytical sample enrolled into CLoCk 12-months post index-PCR test, we actively chose not to focus on comparisons using retrospectively reported data (e.g., symptoms at time of testing and pre-testing conditions)."

4. Line 188 – What does 'examining the data visually' actually mean (ie did you investigate further only the ones that looked obviously different)?

Response: We did not do any further investigation. We have edited to state (page 10 lines 200-201) "We also examined the prevalence of individual symptoms graphically."

5. The NP and PP groups were not used in presenting many of the PCC results because, the date of the infection after baseline was not asked so investigators could not guarantee that the symptoms reported at 24mo were due to PCC or an infection happening in the last month (line 193-196). Was any sort of sensitivity analyses conducted for CYP that had dates of their PCR tests recorded in the database (as opposed to those you only reported a home lateral test), to see how the results would have changed if these cases had been included? It was a huge loss of data to not include these groups and presumably, the vast majority of those reporting an infection after baseline were not infected/reinfected in the 3 months prior to taking their 24mo survey.

Response: Thank you for raising this. Based on this reviewer's comment and those from reviewer 1 we have now edited the text and Table 3 to include the PP and NP groups (along with the NN and PN groups). The results (page 12 lines 251-253) now state: "Approximately one-quarter of CYP fulfilled the broad Delphi research definition of PCC 24-months post index-testing in the NN, PN and NP groups (Table 3); while 30% of the PP group met this broad PCC definition 24-months after their first infection." We have also been explicit in the Statistical methods (page 10 lines 205-206), explaining the underlying assumptions for PCC in the PP and NP groups: "Hence, for these two groups, we assume that their last infection was over 3-months prior to the 24-month questionnaire." In the discussion (pages 19-20 436-438) we also state "We acknowledge that some of our estimated prevalence of PCC in the NP and PP groups could be due to acute infection because we are uncertain of the timing of the (re)infection."

Discussion & General Comments/Edits

6. Data related to vaccination status is not consistent with multiple studies in adults. Do the authors think this is due to limitations in the study design (unable to determine temporality due to self-reported test dates (line 200), biological reasons, or both? It also appears from the data that the vaccination rates are much higher in the NN group than the PP group (lines 217-218) which also report the lowest and highest levels of PCC correspondingly (there also seems to be differences in Sup Table 2 on number of symptoms and number of vaccine doses). Was this considered in any sort of multi-variable modelling? Can data be presented on vaccination status at initial baseline testing? This could at least establish who was vaccinated/unvaccinated at initial infection and be able to compare PCC status at 24mo. It also seems possible to examine temporality amongst the subset of participants with complete follow-up.

Response:

We are not sure what the reviewer means by "Data related to vaccination status is not consistent with multiple studies in adults". Does the reviewer mean that since vaccination has been well established to prevent SARS-CoV-2 infection in children and adults, one would anticipate that if the vaccine stops an individual getting infected, then the individual would not subsequently develop long COVID. We do not disagree.

The more difficult question relates to secondary prevention – does vaccination in individuals who are already suffering long COVID ameliorate their symptoms? We are aware of the following analysis in adults which suggests not only primary prevention but also that secondary mitigation is plausible: Ayoubkhani D, Bermingham C, Pouwels KB, et al. Trajectory of long covid symptoms after covid-19 vaccination: community based cohort study. BMJ 2022;377:e069676. For pre-existing long COVID in adults, vaccination was associated with a 12.8% decrease in self-reported prevalence of long COVID; a second dose was associated with an 8.8% decrease.

However, we are not aware of any publication relating to children which addresses this question. We would therefore have liked to address this but because we don't know when vaccination occurred in relation to infection/re-infection we are unable to have confidence in the suggested analysis. And as the reviewer notes, we acknowledge the limitation in the study design that we are unable to determine temporality due to self-reported tests.

With regard to the comment "Can data be presented on vaccination status at initial baseline testing? This could at least establish who was vaccinated/unvaccinated at initial infection and be able to compare PCC status at 24mo", a COVID-19 vaccine (the Pfizer m-RNA vaccine) was first offered to all young people aged 16 to 17 in England from 23 August 2021 and to school children aged 12 to 15 from 20 September 2021." We are confident that vaccination happened after the initial infection as the Index test occurred between September 2020 and March 2021 whereas the vaccine rollout to young people occurred in August 2021. We have now added a statement to that effect in the Methods (page 8, line 155-157). We did compare all participants who had Long COVID before August 2021 (from first contact at 3m or 6m) with those after August 2021 (from first contact at 12m) as below

Long COVID before Aug 2021 (from first contact at 3m or 6m): index +ves: **25.08%** (2108/8405) index – ves: **18.82%** (1784/9481)

Long COVID after Aug 2021 (from first contact at 12m): index +ves: **26.64%** (1408/3877) index – ves: **23.22%** (1821/7841)

However, we don't feel that this analysis is particularly meaningful given the very large number of variables other than vaccination that could have influenced Long COVID including background infection rates, school closures and type of variant.

7. There seems to be a slight disconnect for group NN throughout the paper. While they are presumably the 'negatives' the assumption throughout the paper (not just what was noted in the discussion) seems to be that the NN group represents children who just never tested positive vs. those who have truly never been infected. While this is certainly likely, it may be good to set this up more in the introduction.

Response: We have edited the entire manuscript to ensure that the NN group is referred to as the group of CYP that have never tested positive for SARS-CoV-2. We added to the introduction (page 7 lines 127-130): "Instead, here we report on symptoms self-reported 24-months post-testing on over 12,600 CYP categorized into four groups by infection status over the 24-month period, including a group that never tested positive (which includes true 'negatives' and/or undetected infections)." We have also edited the discussion to emphasise this point (page 19 lines 420-423): "For example, some CYP categorised as never testing positive for SARS-CoV-2 infection may not be true 'negatives' (i.e., NN). Thus, reported differences between the NN group and the infected groups (PP, PN and NP) are conservative and might be larger if we had a confirmed group of true 'negatives'."

8. Line 301 – compared, not compare

Response: This has been corrected.

Tables and Figures:

9. Table 1 – Were there any significant differences in responders vs. non-responders? If so, they should be noted.

Response: We have added this into Table 1 and reference these differences in the first paragraph of the Results (page 11).

10. Table 3 – The NN and PP groups look almost identical for all the standardized scoring (and also close on the prevalence of PCC). Since these standardized surveys were in use before the pandemic, did the group ever compare the NN groups scores to any available for the same age group before the pandemic? It might give an indication of what may be due to undocumented infection/stress of being a kid during a pandemic and what is just general CYP stress or ailments.

Response: We have added the following text with references to compare with pre-pandemic data (page 16 lines 351-359): "The types of symptoms reported are known to be prevalent in the general adolescent population, raising questions about the role of the initial SARS-CoV-2 infection in their aetiology^{23,24}. For example, those who did not fulfil the PCC Delphi research definition, had SDQ scores broadly in-line with normative data from 11-to-15-year-old British CYP pre-pandemic (i.e. mean (SD): hyperactivity=3.8(2.2); emotional symptoms=2.8 (2.1); peer problems=1.5(1.4); conduct problems=2.2(1.7))²⁵. Likewise, the SWEMWBS score among those who did not fulfil the PCC Delphi

research definition were similar to those reported pre-pandemic in 10-to-16-year-old Danish school children (mean=23.33 (SD=3.78))²⁶."

REVIEWERS' COMMENTS:

Reviewer #1 (Remarks to the Author):

This revision is excellent, well written, flows nicely, and the messaging is clear. The authors have appropriately addressed all my previous concerns.

Reviewer #2 (Remarks to the Author):

Thank you for the thorough response to the edits and comments. I have no further edits or suggestions and would recommend publication.